# Evaluating Seed Enhancement Technology’s Effects on Seed Viability during Multi-Year Storage: A Case Study Using Herbicide Protection Pellets

**DOI:** 10.3390/plants12203662

**Published:** 2023-10-23

**Authors:** Owen Baughman, Anna Hosford, Emily Ralston

**Affiliations:** The Nature Conservancy, 67826A Hwy. 205, Burns, OR 97720, USA

**Keywords:** ecosystem restoration, seed storage, activated carbon, sagebrush, perennial grass, germinability

## Abstract

The viability of seed often decreases during multi-year storage. For seed enhancement technologies (SETs) that apply treatments to native seed prior to sowing in restoration projects, it is important to determine if SETs affect the rate of viability loss in storage to understand if treated seeds can tolerate storage or if they must be sown immediately after treatment. Examining herbicide protection pellet (HPP) seed technology, we conducted germination trials on 10 seedlots of four species to compare three treatments: original bare seed kept in seed storage for 2–3 years, seed retrieved from 2–3-year-old HPPs made from pre-storage original bare seed (old HPPs), and seed retrieved from HPPs that were freshly-made using post-storage original bare seed (new HPPs). For three perennial bunchgrasses, we saw equal or higher germinability of seed from old HPPs compared to the original bare seed and new HPPs, suggesting application of HPP technology to these species prior to multi-year storage is suitable. For the seeds of a perennial shrub, although we saw greater germination of original bare seeds compared to old HPPs, the lowest germination was from new HPPs, still suggesting HPP application prior to storage as a suitable practice. We suggest these tests be performed with all new SETs under development for ecological restoration.

## 1. Introduction

The most well-planned native plant restoration efforts will involve deploying the highest quality, genetically appropriate seed at the best time [1,2]. Obtaining that seed and maintaining its quality until deployment is a key challenge [3,4]. All seedlots eventually lose viability in storage over time, even under ideal storage conditions [5], and genetically diverse seedlots of native species arising from natural populations present additional challenges to viability maintenance during storage [6]. Ideally, seed is used as soon as possible to avoid unnecessary wastage caused by storage, but the complexities of native seed markets and unpredictability of restoration demand are increasingly requiring strategic pre-planning and storage of seed ahead of unplanned, sudden needs [1,4,7].

Seed enhancement technologies (SETs) are an increasingly popular research topic for seed-based restoration of native plants. Broadly defined, as they pertain to native species, SETs are any treatment applied to seed prior to sowing with the intent of improving restoration efficiency and outcomes by alleviating the effects of specific barriers to successful seed delivery, germination, emergence, or establishment [8,9]. When applied at scale for ecosystem restoration, the use of SETs will unavoidably result in additional steps and logistics in existing seed supply chains, and this is likely to increase the time between seed harvest and seed sowing. Whether the application of SETs affects seed viability in storage has not been widely assessed, but it is crucial to understanding when to apply treatments in the seed supply chain to prevent losses in seedlot quality. For example, although seed priming of food crops such as lettuce, tomato, corn, and rice improves the performance of seedlots, the notably reduced longevity of primed seeds under standard storage conditions requires mitigation via expensive cold storage or creates unavoidable logistical complications for the timing of the treatment with regard to sowing [10,11].

Understanding whether new SETs will affect the storage life of the seeds they contain is a critical component in evaluating their full efficacy as realistic, scalable solutions to restoration challenges. Here, we use one SET currently under development for use in native plant restoration as a case study to demonstrate one simple approach to assessing treatment effects on seed viability in storage.

Herbicide protection seed treatment is a SET being developed to protect seed from the toxic effects of pre-emergent herbicides frequently used to control exotic invasive annuals. This treatment is designed to allow seedling establishment during the period of maximum herbicide efficacy and, thus, lowest competition with targeted weeds [12,13]. Production of this technology involves combining seed with activated carbon and other binders into either multi-seed pellets via a wet extrusion pelleting process identical to the production of pasta [13,14,15], or into single-seed coatings [16,17,18] via rotary or pan coating techniques commonly used in the application of other seed coatings [9].

It is unknown if seed encased in herbicide protection technologies lose viability in storage at different rates than bare seed. It is plausible that the long-term viability of a seedlot could be altered by the processes used to adhere carbon to the seed, either from exposure of seeds to short periods of increased temperature and moisture, or the chemical properties of the carbon or binders involved, or both. Addressing this question will inform decisions about whether the application of the SET to the seeds must be performed immediately prior to sowing or whether it can be performed after seed cleaning, but prior to storage by the end user. In regions where multi-year storage of seed is common or lead times on filling seed orders for restoration are often short, the ability to have seed treated with SETs prior to storage would avoid added logistical barriers to their use in restoration.

Using multiple seedlots of four perennial species native to North America’s sagebrush (*Artemisia* spp.) steppe that had been used to produce multi-seed herbicide protection-extruded pellets (hereafter, HPPs), we asked (1) which seed treatment maintains the highest germinability after 2–3 years of controlled storage: seed within HPPs made prior to storage, seed within HPPs that were made after seed storage, or untreated bare seed, and (2) do different storage times (two vs three years) modify germinability of seed stored in HPPs?

## 2. Results

The percentage of seeds that germinated was affected by seed treatment for some species but not others (Figure 1). For *E. elymoides*, germination of old HPPs (46%) was higher than bare seed (35%), but with new HPPs, no different from either (37%) (seed treatment main effect; F_2,67_ = 3.72, *p* = 0.029). For *P. spicata*, germination of new HPPs (36%) was lower than old HPPs and bare seed (both 56%), which did not differ from one another (seed treatment main effect; F_2,44_ = 20.8, *p* < 0.001). For *P. secunda*, there was no effect of seed treatment (seed treatment main effect; F_2,67_ = 3.12, *p* = 0.051). For *A. tridentata*, the germination of all three seed treatments varied from one another, with bare seed having the highest (79%), new HPPs the lowest (20%), and old HPPs an intermediate germination (59%) (seed treatment main effect; F_2,44_ = 58, *p* < 0.001).

In the second model investigating whether storage time of old HPPs was a confounding factor for *E. elymoides* and *P. secunda*, there was no effect of species, storage time, or their interaction on the percentage of seeds that germinated (F_1,3–57_ = 0.52–1.94, *p* = 0.23–0.54; not shown).

## 3. Discussion

A diverse array of new and developing seed enhancement technologies (SETs) promise to improve success rates of seed-based restoration efforts using native species [8,9] but would also add additional steps and logistical concerns to seed supply chains. Before considering mass-production and scale-up of any SET, it is crucial to understand when to apply the SET to, avoid negative impacts on seed quality, especially if one or more years of seed storage are common or expected. Our case study focused on herbicide protection pellet (HPP) seed technology, which combines native seeds with activated carbon to protect them from the deleterious effects of pre-emergent herbicides regularly used in restoration sites to control invasive annual grasses.

For all three perennial grass species tested, storing seed for two or three years as fully prepared HPPs resulted in no reductions in germinability when compared to seed stored for the same amount of time as bare seed. These results suggest that the application of HPP technology to seed prior to multi-year storage in constant, favorable conditions (15–19 °C and 25–35% RH) is unlikely to impact the germinability of perennial grass seed. Additionally, none of the four species tested showed higher germination of new HPPs made after storage compared to old HPPs made prior to storage, and *P. spiciata* and *A. tridentata* in fact demonstrated lower germination of new HPPs than old HPPs. We also found no evidence that different storage times (two vs three years) modified our results, though our test of this relied on a more limited number of seedlots. Together, our findings suggest that the ideal application time of this technology is to bare seed prior to multi-year storage rather than after, at least for the storage times tested (up to three years).

Despite the important limitations of our study (below), the implications of our findings are that at least one formulation of herbicide protection seed enhancement technology, HPP, is compatible with a timeline in which the SET is applied to the seed prior to multi-year storage, and it likely does not require production immediately before sowing to maintain seed quality. This compatibility is important because, for example, the majority of seed-based restoration in North America’s sagebrush steppe region is planned and performed in short periods (2–5 months) after unplanned wildfires occur, and it is already challenging to coordinate and obtain the desired amount of the correct seed in such a short window [1,2]. Any SET that requires its application to occur just prior to sowing would significantly limit, if not prevent, its adoption into such restoration timelines for logistical reasons, even if it were an effective treatment.

In an investigation of dormancy alleviating seed treatments, Turner et al. [19] found that applying treatments prior to storage had no disadvantage compared to applying them after storage. The authors noted that applying the treatments before storage was a significant advantage over the “double handling” that would be required to apply them just prior to sowing. We surprisingly found no other recent studies on the development of SETs for native species restoration that assessed or addressed the ideal sequence of SET application with respect to multi-year storage, and only Anderson et al. [20] highlighted the need to investigate the implications of their seed conglomerating treatment on long-term viability in storage.

For one species, we observed some negative effects of SETs on seed viability in storage but also some evidence for other, positive implications. The seed of *A. tridentata* germinated better after storage as bare seed than when made into and recovered from new HPPs or recovered from old HPPs produced before storage, though old HPPs had the smallest decline in germinability. This generally supports the findings of prior studies incorporating *A. tridentata* seeds into HPPs, which have also observed overall negative effects of the technology on this small-seeded, perennial shrub species [18,21]. Interestingly, our findings imply that even though storage of bare seed in refrigerated conditions is recommended for this species and resulted in the maintenance of relatively high germinability of bare seed (75%), this may not be needed once the seed is made into pellets. The germinability of seed in old HPPs stored for multiple years in non-refrigerated storage settings was higher than the germinability of seed recovered from new HPPs that received multi-year refrigerated storage as bare seed prior to pellet production. If future refinements to herbicide protection treatments of this species can eliminate the negative impacts on germination, our results hint that the pre-storage application of the technology could eliminate or reduce the need for costly refrigerator or freezer storage for this species. Additional trials with more seedlots, multiple storage temperature treatments, and repeated germinability assessment over time would be needed to confirm this notion.

We recognize that our relatively small dataset was limited to only four species and a narrow range of storage times, and it did not evaluate other potentially important factors that varied across our seedlots and among species, such as maternal environments of seed, original seed age, seed cleaning procedures, and pre-purchase storage conditions [4,22]. Additionally, our assessment involved only one set of intentionally ideal storage conditions for each seedlot, and our results may have differed with different or less-controlled storage conditions. However, because our main finding that pre-storage application of HPP technology was either no different from, or more beneficial than, post-storage application was consistent across all species tested, we believe this finding is relatively robust and is a promising starting point for additional iterations of this technology. Similarly, although it appears that we observed differences among species in absolute germination rates and/or the effect of HPP treatments on germinability, we do not suggest that the observed differences can be attributed to species alone, as they are likely influenced to some degree by variations in these and other untested factors.

The future of HPP technology as an at-scale tool for restoration of many species is unclear, with additional trials and refinements still needed [8,12,18]. Additionally, another method to apply activated carbon to seeds, namely seed coating, is under investigation [16,17,18,23]. The results of the current study with extruded pellets should not be expected to hold for all new herbicide protection technologies and formulations. However, future refinements to herbicide protection technologies should consider the current results as a useful target, as they demonstrate that at least one formulation of this technology (extruded pellet) can be applied with minimal effects to the germinability of perennial grasses after multi-year storage. Of course, it should be noted that our study only assessed seed germinability after manual recovery from HPPs, and that laboratory and field tests of post-storage seed emergence from within the finished pellets or coatings is a separate and perhaps more important test, which may yield different results. Future efforts to assess the shelf life of SETs should consider both types of tests.

We strongly recommend that the development and assessment of any other SETs for use with native species for ecosystem restoration should include, at minimum, the relatively simple tests conducted here. To do this, we recommend that researchers developing SETs place stock seed and SET prototype samples into realistic but controlled multi-year storage settings, as we have done, or investigate and perform accelerated aging tests [24,25]. This knowledge of shelf life is critically needed in order to foresee any additional complications that seed treatments may add to the already complicated native seed supply and delivery chain [1,3], and to produce suggested timelines for their application with respect to seed storage.

## 4. Materials and Methods

We used seed of four species native to western North America’s sagebrush steppe ecosystem (Table 1): one perennial shrub, *Artemisia tridentata* var. *wyomingensis* (Wyoming big sagebrush), and three cool-season perennial grasses, *Elymus elymoides* (Bottlebrush squirreltail), *Poa secunda* (Sandberg bluegrass), and *Pseudoroegneria spicata* (Bluebunch wheatgrass). All four species are non-dormant, requiring no pre-treatments to improve germination, and germinate readily at a range of mild temperatures when wetted [26]. This experiment utilized seed and HPPs left over from multiple prior experiments over multiple years (hereafter, “old HPPs”). All bare seed and old HPPs had been stored in non-airtight plastic mesh grain bags in the same location, a dedicated seed-storage facility that maintained conditions of 15–19 °C and 25–35% relative humidity (RH), except for both *A. tridentata* bare seed samples, which were stored as recommended in air-tight plastic bags in a refrigerator (1–4 °C, see below). We identified 10 seedlots of the four species (Table 1) for which we had both (1) leftover bare seed and (2) HPPs that had been produced two or three years prior to those same lots of bare seed. Two seedlots (EL2/3 and PO2/3; Table 1) had been made into HPPs over several years, meaning we had old HPPs with multiple storage times for these seedlots. See below for how these specific seedlots were included in the analyses.

For each seedlot, the same recipes, methods, and machines used to produce the old HPPs were used to produce fresh batches of HPPs (henceforth “new HPPs”) just prior to conducting germination tests. Production of HPPs involved (1) preparing a dry mixture of 33–34% (by weight) Darco Gro-safe activated carbon (Norit Activated Carbon, Marshall, TX, USA), 42–44% Pelbon bentonite clay (American Colloid Company, Bethlehem, PA, USA), 13–14% sieved (1.9 mm) compost fines (Deschutes Recycling, Bend, OR, USA), 6% sieved (1.9 mm), Worm Gold worm casting fines (California Vermiculture, Cardiff, CA, USA), and the desired amount of dry seeds, (2) mixing with 0.647 mL of water per g dry material in a planetary mixer for 90 s, (3) extruding the wet mixture in a TR110 pasta extruder (Rosito Bisani, Los Angeles, CA, USA) fitted with Teflon dies perforated with round extrusion channels of either 4.8 mm (*A. tridentata*) or 8 mm (all other species) in diameter, (4) cutting extruded material to a length that is approximately twice the diameter using the onboard cutting knife, and (5) immediately drying extruded pellets on mesh screens over forced air at 24–38 °C until dry (15 min at 65 °C, then 60–240 min at 24 °C). Additional production details are described by Brown et al. [15] and Baughman et al. [18]. This resulted in three treatments for each seedlot: (1) bare seed (control), (2) old HPPs, and (3) new HPPs.

We conducted a germination trial on blotter paper in petri dishes in Hoffman SG2-22 germination chambers with a 12-h photoperiod under 5000k, 1232 lumen LED lights, and a 20 °C/10 °C diurnal cycle. Dry bare seed were plated directly into petri dishes on blotter paper saturated with water, with 25 seeds per dish. For HPP treatments, a single dry sample of HPPs containing approximately 25 seeds, based on recipe specifications, was processed separately for each dish by soaking HPPs in tap water for no more than 10 min, then sieving over a 0.7 mm (*E. elymoides* and *P. spicata*), 0.5 mm (*P. secunda*), or 0.35 mm (*A. tridentata*) sieve with gentle agitation in water. All seed recovered in the sieving (even if over 25 seeds) were then plated into a single dish. This removal process was performed gently to avoid any seed damage. Eight replicate dishes were prepared for each combination of seedlot, species, and seed treatment. Dishes were kept moist via watering every two to four days, and germinated seed was counted and removed every two days until the entire eight-dish set showed no new germination for longer than seven days.

The single response variable was the percentage of whole seeds that germinated (%). Two linear mixed-effects models were conducted using JMP 15 (SAS Institute; Carey, NC, USA). The first, main model, included seed treatment (bare seed, old HPP, new HPP) as a fixed effect and seedlot (nested within species) as a random effect. This model was run separately for each of the four species (*A. tridentata*, *E. elymoides*, *P. secunda*, *P. spicata*) because species was a significant and interactive factor when included in a prior, factorial model with seed treatment. For seedlots EL2/3 and PO2/3, which had samples of old HPPs of both two and three years of storage time available, only the three-year-old samples were included in the main model. A second model was performed to investigate whether storage time of old HPPs had an effect on germinability (a potentially confounding factor not included in the main model for *E. elymoides* and *P. secunda*). The second model included two fixed factors: species (*E. elymoides*, *P. secunda*) and old HPP storage time (2 vs. 3 years), and their interaction, and seedlot (nested within species) was included as a random effect. This second model included only the old HPP treatment (omitting bare seed and new HPP) from six seedlots, including old HPPs of both storage ages (two and three years) for seedlots EL2/3 and PO2/3. For both models, significance was evaluated at the *p* < 0.05 level, and post-hoc Tukey HSD tests were performed to determine differences among least squares means. The random factor seedlot was included in all models as a means to account for differences among the seedlots in untested factors such as viability, purity, original seed age, and maternal environment. There was no significant effect of the random factor seedlot for any species in any model (Wald *p* = 0.166–0.55), so the results reflect data pooled by seedlot within each species.

## 5. Conclusions

New seed enhancement technologies for native plant restoration add additional steps to seed supply chains, and their application should be timed to minimize negative effects on seed quality. Multi-year storage of seed is common in some regions, and loss of seed germinability in storage is a common concern. Our findings suggest that applying one seed enhancement technology (herbicide protection pellets) prior to multi-year storage rather than after had the fewest impacts on final seed viability, and this information is critical to understanding how this new technology could fit into already challenging seed supply chains. Tests to confirm if and how seed treatments affect viability loss in storage are recommended for other seed enhancement technologies being developed and refined.

## Figures and Tables

**Figure 1 plants-12-03662-f001:**
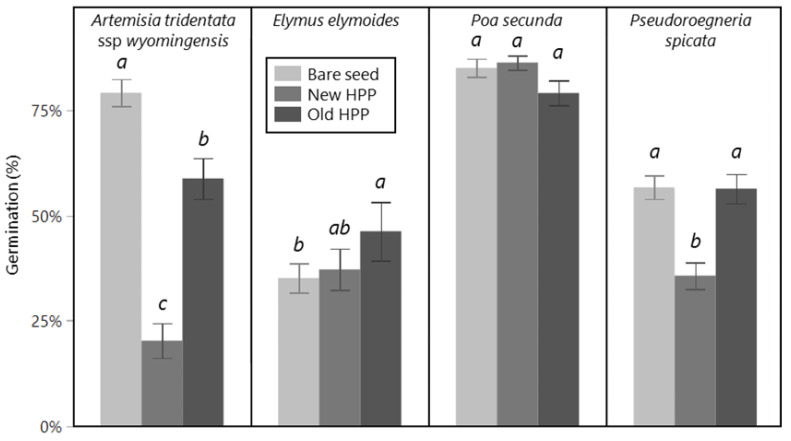
Percent germination in petri dish trial of bare seed (light gray), newly made HPPs (medium gray), and older HPPs (dark gray) separated by species (across the top). Note that the seed used in the bare seed and new HPP treatments was stored as bare seed until the new HPPs were made immediately prior to the germination trial, whereas seed in old HPPs was stored as fully prepared HPPs. Storage conditions for all HPPs and bare seed of all species were in non-airtight paper bags or plastic grain bags at 10–20 °C, 25–35% RH, except *A. tridentata* bare seed (which includes the seed made into new HPPs), which was stored in air-tight plastic bags at 1–4 °C, 20–30% RH. Within each panel (species), means are significantly different from one another at the *p* < 0.05 level, unless displaying the same letter, based on Tukey HSD tests. For each bar, errors are one standard error, and n = 16 (*A. tridentata*, *P. spicata*) or n = 24 (*E. elymoides*, *P. secunda*).

**Table 1 plants-12-03662-t001:** Seedlot and seed treatment details for the 10 seedlots used in the experiment.

Seedlot ID	Seed Source Name and Origin	Old HPP Age (Years)	Pellet Diameter (mm)
*Artemisia tridentata wyomingensis*			
AR-3	“Winnemucca”; Nevada, USA	2	4.8 mm
AR-4	“Vine Hills”; Oregon, USA	2	4.8 mm
*Elymus elymoides*			
EL-1	n/a; Oregon, USA	3	8 mm
EL-2/3	“Sonoma”; Nevada, USA	2, 3	8 mm
EL-4	“Deschutes”; Oregon, USA	2	8 mm
*Poa secunda*			
PO-1	“Paulina”; Oregon, USA	3	8 mm
PO-2/3	“Panther”; Nevada, USA	2, 3	8 mm
PO-4	“Mt. Home”; Idaho, USA	2	8 mm
*Pseudoroegneria spicata*			
PS-1	“John Day”; Oregon, USA	3	8 mm
PS-2	“Crooked River”; Oregon, USA	3	8 mm

## Data Availability

The data supporting this work are openly available from DataDryad.org and will be assigned the following DOI upon publication: DOI:10.5061/dryad.95x69p8qs.

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
