# Peer review of "Evaluating Seed Enhancement Technology’s Effects on Seed Viability during Multi-Year Storage: A Case Study Using Herbicide Protection Pellets"

_plants, 2023, doi:10.3390/plants12203662_

Round 1
Reviewer 1 Report
Introduction:
The authors do a good job of summarizing the current state of knowledge on the impact of seed enhancement technologies (SETs) on seed viability in storage.
Here are some specific comments that may to consider:
- The authors could provide more information about the different types of SETs and how they work. This would help the reader to better understand the potential impact of SETs on seed viability
Material and method
Overall, section effectively outlines the experimental procedure and method used in the present study.
Results
- The results section presents the study was well-designed and the results are clear and concise. The authors have done a good job of controlling for potential confounding factors, such as the seed lot. Overall, the study provides valuable insights into the effect of seed treatment and storage time on the germination of seeds. The results suggest that the best seed treatment for a particular species may need to be determined empirically, and that old HPPs can be stored for a period of time without affecting their viability.
- Here are some specific comments that I would like to suggest…
- The authors could also provide more information about the storage conditions that were used in the studies. This would help the reader to better understand the results of these studies
- The study was conducted under controlled laboratory conditions. Is it possible that the results would be different under field conditions?
- Discussion:
Overall, the discussion section is well-written and informative. The authors do a good job of summarizing the results of their study and discussing the implications of their findings. However, the authors could provide more information about the different types of SETs that are currently being developed, and the implications of their findings for the use of HPP technology in native plant restoration.
Here are some specific comments that you may want to consider:
- The authors could provide more information about the different types of SETs that are currently being developed. This would help the reader to better understand the potential implications of the authors' findings.
- The authors could also discuss the limitations of their study. For example, they only tested four species of plants, and they only tested storage times of up to three years. It is possible that the results of the study would be different for other species or for longer storage times?
- The authors could also discuss the implications of their findings for the use of HPP technology in native plant restoration. For example, do the authors' findings suggest that HPP technology should be used to store seed for all native plant species? Or are there some species for which HPP technology is not suitable?
- References:
The references are properly formatted in terms of citation style. The authors have done a good job of citing relevant literature to support their claims.
Reviewer 2 Report
See pdf file.

Reviewer 3 Report
1. IN Materials and Methods. herbicide protection pellet method needs to be described in detail,in old HPPs and HPPs. What are the differences between the two methods and what improvements have been made?
2. The result is too simple, just a table.
3. The experimental materials, methods and results are too simple and need to be described in detail
4. The title of the paper is unclear and should be revised
Moderate editing of English language required.
Reviewer 4 Report
The manuscript “Should Seed Enhancement Technologies for Native Plant Restoration be Applied before or after Seed Storage? A Case Study using Herbicide Protection Pellets” is dealing with seed viability after multi-year storage. Authors examined the effects of herbicide protection pellet (HPP) seed technology to compare germination capacity of bare seeds kept in seed storage for 2-3 years, seeds retrieved from 2–3-year-old HPPs and seeds retrieved from HPPs that were freshly-made using post-storage original bare seed (new HPPs). The examination was done with four plant species native to western North America’s sagebrush steppe: perennial shrub, Artemisia tridentata var. wyomingensis, and three cool-season perennial grasses, Elymus elymoides, Poa secunda, Pseudoroegneria spicata. Authors found that the best timing of application for seed enhancement was prior, rather than after multi-year storage especially for three perennial bunchgrasses, suggesting application of HPP technology to these species prior to multi-year storage is suitable. These results has some merit and manuscript falls in a scope of the journal.
Manuscript is very well written and can be accepted for publication in journal Plants after minor corrections, which are highlighted in text of manuscript and explained to authors.
Line 66 Is “?” necessary in the middle of the sentence?
Line 242, 242. What means this sentence? It looks like it remained as suggestion from previous versions of the manuscript
Lines 244-274, 293-303. These sections remained from Template and should be deleted since all relevant information data are reported between.

Round 2
Reviewer 3 Report
Minor editing of English language required.
The author has answered the questions I raised and revised it.